# Systematic Physical Exercise and *Spirulina maxima* Supplementation Improve Body Composition, Cardiorespiratory Fitness, and Blood Lipid Profile: Correlations of a Randomized Double-Blind Controlled Trial

**DOI:** 10.3390/antiox8110507

**Published:** 2019-10-23

**Authors:** Marco Antonio Hernández-Lepe, Francisco Javier Olivas-Aguirre, Luis Mario Gómez-Miranda, Rosa Patricia Hernández-Torres, José de Jesús Manríquez-Torres, Arnulfo Ramos-Jiménez

**Affiliations:** 1Medical and Psychology School, Autonomous University of Baja California, Tijuana 22390, Mexico; marco.antonio.hernandez.lepe@uabc.edu.mx (M.A.H.-L.); jose.de.jesus.manriquez.torres@uabc.edu.mx (J.d.J.M.-T.); 2Health Sciences Department, University of Sonora, Cajeme 85010, Mexico; francisco.olivas@unison.mx; 3Sports School, Autonomous University of Baja California, Tijuana 22390, Mexico; lgomez8@uabc.edu.mx; 4Physical Culture Sciences School, Autonomous University of Chihuahua, Chihuahua 32310, Mexico; rphernant@yahoo.com; 5Biomedical Sciences Institute, Autonomous University of Ciudad Juarez, Ciudad Juárez 32310, Mexico

**Keywords:** nutraceuticals, antioxidants, *Arthrospira maxima*, physical exercise, obesity, randomized trial

## Abstract

Cardiovascular diseases are part of the highly preventable chronic diseases associated with changes in lifestyle. Within them, physical activity, low-fat and high-fiber diets are distinguished as the main support for prevention, even when supplementation with nutraceuticals has become a very common practice. Fifty-two young sedentary men with excess body weight (body mass index (BMI) ≥ 25 kg·m^−2^) were enrolled in a randomized-crossover controlled trial [six weeks of a systematic physical exercise with *Spirulina maxima* or placebo supplementation (4.5 g·day^−1^)]. Body composition, blood lipid profile, and maximal oxygen uptake were determined pre/post intervention. Pairwise comparisons showed a significant improvement (*p* < 0.01) on blood lipid profile in the group of exercise plus *Spirulina maxima*. Moreover, correlations of absolute changes of BMI, body fat percentage, blood lipids and maximal oxygen uptake were statistically significant (*p* < 0.01). These results indicate that the *Spirulina maxima* supplementation could be acting in a synergistic way with exercise due to the enhanced effects on body composition, cardiorespiratory fitness, and blood lipid profile. This phenomenon should be considered to reduce risk of cardiovascular disorders. Trial registration: Clinical Trials, NCT02837666 (July 19, 2016).

## 1. Introduction

Dyslipidemia is a major risk factor for metabolic syndrome, cardiovascular diseases, and type 2 diabetes mellitus, is characterized by elevated blood levels of total cholesterol (TC), triacylglycerols (TAG), and low-density lipoprotein-cholesterol (LDL-C), and low levels of high-density lipoprotein-cholesterol (HDL-C). TC, TAG, and LDL-C often increase directly with body mass and central adiposity [1]. In particular, in young adults, adiposity-dyslipidemia follow a relationship (*R*^2^ = 0.11−0.17) with body mass index (BMI) with a sharply increment between 25–30 kg·m^−2^ [2]. Since overweight and obesity are growing exponentially in young adults, there is an urgent need for effective preventive/control measures to stop this pandemic consortium (obesity-dyslipidemia) [3].

To promote cardiovascular health, lifestyle changes must be considered as guideline to reduce people’s sedentarism and to boost healthy-eating patterns. It is well known that sedentarism and energy-dense diets reduce insulin sensitivity and fat oxidation that leads to dyslipidemia. Conversely, systematic exercise increases the maximal oxygen uptake (VO_2_max), promotes energy expenditure from fat stores [4,5] and improves the plasma lipidome. Also, a heart-healthy diet that include functional nutrients and phytochemical such as plant and marine proteins, n3- fatty acids, phytosterols and polyphenols (among others), seek to reduce the intake of saturated/trans-fatty-acids and cholesterol, and their unfavorable metabolic consequences [6]. 

In this sense, several research communications provide a substantial body of evidence on the benefits of supplementing *Spirulina* sp. This group of filamentous, spiral-shaped, blue-green cyanobacteria improves several cardio-metabolic biomarkers such as TC, TAG, and LDL-C, fasting blood glucose and diastolic blood pressure, without side-effects in humans [7,8,9] and animal models [10,11,12]. Such benefits seem to rely on several antioxidant compounds (e.g., C-phycocyanin) and enzyme inhibitors/activators [7,8,13]. We hypothesized that a systematic physical exercise program and *Spirulina maxima* (*S. maxima*) intake in overweight and obese men has beneficial effects on the body composition, blood lipid profile and cardiorespiratory fitness, and those effects are highly correlated.

## 2. Materials and Methods 

### 2.1. Supplements Characteristics

*Spirulina maxima* was purchased from Alimentos Esenciales para la Humanidad S.A. de C.V. (Mexico City, Mexico). Its chemical/functional composition and biosafety were evaluated before use [14]. Placebo composition consisted of a low-calorie saccharine powder. Both supplements were encapsulated in dark capsules (0.5 g each one) to present the same organoleptic characteristics, and subjects were asked to ingest three capsules before each daily meal. Due the lack of information in the literature, dose (4.5 g·day^−1^), duration for the treatment period (six weeks) and the washing period (two weeks) were considered long enough according to a systematic review of clinical trials that used Spirulina as treatment [9].

### 2.2. Ethics Approval and Consent to Participate

The Consolidated Standards of Reporting Trials (CONSORT) checklist (Appendix A) and flow diagram (Appendix A) are available as Appendix A. The study protocol was approved by the Autonomous University of Ciudad Juarez (UACJ) review board and carried out following the declaration of Helsinki, and the trial was registered at clinicaltrials.gov with trial ID: NCT02837666 (Hypolipidemic and Antioxidant Capacity of Spirulina and Exercise, https://clinicaltrials.gov/ct2/show/NCT02837666?cond=Hypolipidemic+and+Antioxidant+Capacity+of+Spirulina+and+Exercise&rank=1). Written informed consent was obtained from all participants prior to any study-related procedures.

### 2.3. Inclusion and Exclusion Criteria

Male volunteer, sedentary (daily energy expenditure <4 metabolic equivalents), with body mass index (BMI) ≥ 25 kg·m^−2^ and 18–35 years old were recruited in an intra-school campaign and personalized interviews to ensure eligibility. Exclusion criteria were drinking more of 100 mL of alcohol a week, taking drugs or diet supplements, presenting diabetes or any other chronic disease, and having an impediment to practice regular physical exercise.

### 2.4. Participants and Study Design

Fifty-two subjects were enrolled in a randomized, double-blind, crossover-controlled trial. Subjects were distributed in one of two groups according with their BMI: overweight (*N* = 27) or obesity (*N* = 25). Each subject received randomly 4.5 g·day^−1^ of *S. maxima* or placebo in two 42-day crossover periods (randomly), separated by 14 days (wash-out period) (Figure 1). A detailed protocol of this study, including sample size calculation, details of randomization, and anthropometry have been published previously [14].

### 2.5. Diet

All participants underwent a nutritional survey to establish an isocaloric diet and keep body weight constant during the study. Two trained nutritionists monitored dietary intake using three 24-hour dietary withdrawals on days 0, 42, 56, and 98, and two food frequency questionnaires on days 0 and 98, in order to ensure the independence of both dietary assessment tools. A nutritionist first inspected diet records for missing data (e.g., missing foods or lack of complete responses) and then analyzed the total intake of calories, protein, carbohydrates, and fats (Diet Analysis Plus, ESHA Research, Salem, OR, USA). Finally, compliance with supplement intake (*S. maxima* or placebo) and diet were monitored weekly by scheduled laboratory visits and carried out by trained nutritionists.

### 2.6. Determination of Maximal Oxygen Uptake (VO_2_max) 

To determine VO_2_max, each subject performed a stress tests at maximum intensity at each session (four in total, day 0, day 42, day 56 and day 98) during the trial, each test consisted in measure oxygen uptake and carbon dioxide produced by a gas analyzer (Cortex Metalyzer 3B, Germany) using a cycle ergometer (Monark Ergomedic 828E, Monark Exercise AB, 105 Vansbro, Sweden). Each test commenced with subjects cycling at a cadence greater than 60 rev·min^−1^ during each test, starting with a external resistance of 50–75 W with increments of 15–30 W every three minutes until the subject can no longer pedal more than 40 rev·min^−1^. The test was considered valid when at least three of the following criteria were met: the heart rate was 90–100% of the maximal predicted by age, rates of perceived exertion (RPE) of at least 9 on Borg CR-10 scale, respiratory exchange ratio > 1.1, and VO_2_ plateau (ΔVO_2_ < 2.1 mL·kg^−1^·min^−1^. At the end of each increment load, glucose and blood pressure were determined to check the subject’s health during each test. The laboratory temperature was maintained at ~22 °C and the humidity at ~50%.

### 2.7. Physical Exercise Program

The training program was individual and designed to elicit an improvement in cardiovascular endurance according with the American College of Sports Medicine (ACSM) guidelines for people who are overweight and for obesity [15]. The exercise protocol was five days a week and began with a warm-up exercise (5–10 min), followed by muscular endurance exercise (20–30 min), then 20–30 min of cardiovascular exercise, and finally, five minutes of stretching. Target heart rate for each exercise session was based on a percentage of heart rate reserve (HRR). The difference between the maximal heart rate observed during the stress test at maximal intensity and the heart rate prior the test, measured with subjects resting comfortable for at least 15 min in a supine position (resting heart rate) was considered the HRR. The cardiovascular exercise was administered as follows: Three days at low to moderate intensity (50–80% of their HRR), and two days at high intensity (80–90% of their HRR).

Muscular endurance exercise consisted of working all muscle groups (arms, legs, chest, back, and shoulders) using a medium-resistance repetition protocol, divided in four different exercises for each specific muscle group, doing three sets of 12–16 repetitions [15]. In order to maintain a constant training stimulus, HRR was recalculated every session of stress test at maximal intensity. Subjects performed the physical exercise program in the UACJ gym, always under the technical supervision of a personal trainer.

### 2.8. Blood Sample Analysis 

Each session before the stress test at maximal intensity, fasting blood samples (8 mL, 8–10 h of fasting) of subjects were collected from the antecubital vein into ethylene diamine tetra acetic acid (EDTA) tubes (four in total, day 0, 42, 56 and 98). Samples were centrifugated at 4 °C at 3000× *g* during 20 min to obtain plasma, and the lipid profile (TC, TAG, LDL-C, and HDL-C) levels were analyzed by standard enzymatic procedures (Spinreact, Girona, Spain) with a spectrophotometer (Epoch, Biotek, VT, USA) at 505 nm. Abnormal cutoffs for TC, TAG, LDL-C and HDL-C were >200, >150, >100, and <40 mg·dL^−1^, respectively.

### 2.9. Statistical Analysis

Data distribution normality was examined by the Shapiro-Wilk test, and the homoscedasticity by the Levene test, for each group. Statistical differences before and after each intervention group were compared using paired t-tests. The associations among variables were evaluated by Spearman correlation. *p* < 0.01 was considered statistically significant. All analyses were conducted using the software SPSS 22.0 (SPSS Inc., Chicago, IL, USA) and are shown in the Appendix A.

## 3. Results

According to their baseline characteristics, participants showed a low cardiorespiratory fitness [15], and subjects with obesity (*N* = 25) had abnormal levels of TC, TAG, LDL-C, and HDL-C, and were considered with dyslipidemia (Table 1).

Daily energy intake after the clinical trial showed no statistical differences (*p* < 0.01) for dietary variables at the end of the study compared to those at the beginning (2146 ± 98 kcal·day^−1^ vs. 2054 ± 104 kcal·day^−1^). No adverse effects of *S. maxima* supplementation or dietary intake were reported during the study.

Before the clinical intervention, obese subjects were classified with dyslipidemia according with their basal blood lipids. However, when the intervention concluded, they showed positive changes in all blood lipids measured, BMI, and VO_2_max (Table 2). Increase in HDL-C levels is highly explained by the fatty acids release by other lipoproteins (and its subsequent oxidation in peripherical tissues) and not by their storage. However, HDL-C remained below normal levels (≥40 mg/dL). 

Regarding treatments effect, only the group with exercise and supplementation with *S. maxima* had a decrease in plasma lipids (Table 3, *p* < 0.01). 

According to the correlation analysis (*p* < 0.01), in the exercise and *S. maxima* supplementation group, BMI decrease as VO_2_max increase, TC and LDL-C decrease linearly, finally, while LDL-C levels decrease, HDL-C increases. In the exercise and placebo supplementation group, TC decrease linearly with percentage body fat, TAG, and LDL-C, while BMI and LDL-C had a proportional decrease too. Finally, in the *S. maxima* supplementation without exercise group, VO_2_max increase as BMI and percentage of body fat decrease, TC and TAG decrease linearly (Table 4).

## 4. Discussion

Health benefits associated with synergism between systematic exercise and *Spirulina* supplementation have been scarcely documented. In this sense, this study provides new insights on the synergistic effects of *S. maxima* and a moderate exercise program through correlations among physiological descriptors such as weight loss, body fat loss, cardio-respiratory fitness. It has been proved that *Spirulina* supplementation reduces body fat and plasma concentrations of TC, LDL-C, and TAG while increasing HDL-C, independent of the administered *Spirulina* dose [7,8]. Likewise, we found statistically significant (*p* < 0.01) improvements on the blood lipid profile from the exercise plus *S. maxima* treatment, so this is the first study focus in providing evidence on the correlations of the biological and physiological effects of a short-term exercising protocol and *S. maxima* supplementation in overweight and obese men.

Although Kalafati et al. [16] studied the effect of *S. maxima* supplementation (8 g·day^−2^ for 8 weeks) in 9 trained men, they reported a prolonged time to fatigue, fat oxidation rate, and antioxidant enzymes. Despite the fact that they did not analyze the effect on VO_2_max, the results suggest that it would be improved, but like this study, the absence of convincing studies of *S. maxima* effects in the literature [7] may be attributed to the studies designs, the absence of a physical exercise program, age, fitness level and absence of correlation analyses between variables; in our study, the latter is further supported by the fact that correlations revealed that the hypolipidemic effect of physical exercise plus *S. maxima* with BMI and VO_2_max was evident in excess weight individuals. 

Ergogenic and antioxidant compounds [7,8,13,16] and enzyme inhibitors or activators [17,18] present in *Spirulina* sp. seems to act in concerted action to reduce body fat and plasma lipids. Even if the molecular mechanisms involved are poorly understood, it has been proposed that some molecules such as C-phycocyanin or polyphenols can maintain oxidative species within healthy limits in order to prevent undesirable events (e.g., oxidative stress). This adequate balance translates into the maintenance of essential functions (regulated by reactive oxygen/nitrogen species; RONS) for physical improvement. These functions include the messenger activities of RONS on physiological adaptations such as angiogenesis, insulin sensitivity, hypertrophy, mitochondrial biogenesis, among others that promotes tolerance to future stressors [19]. In this regard, some studies have evaluated the influence of antioxidant supplementation on alterations induced by physical training, the results obtained being extremely variable [20]. For example, it has been reported that supplementation with vitamin C and vitamin E during exercise decrease the endogenous antioxidant response and in other cases it has a null effect on mitochondrial hormesis [21]. Moreover, C-phycocyanin, is an effective radical scavenger, reduced nicotinamide adenine dinucleotide phosphate (NADPH) inhibitor at the transcription level, and post-translational activator of antioxidant enzymes [7,22]. 

Additionally, other bioactive (e.g., glycolipid H-b2, polyphenols) are also effective inhibitors of pancreatic lipase [22], a mechanism surely involved on the effects of *S. maxima* on postprandial hyperlipidemia in young runners previously reported by our group [23]. Nagaoka et al. [24] attributed *Spirulina* hypolipidemic effects to a decreased intestinal absorption of cholesterol since several *Spirulina* bioactive compounds bind to bile acids at the jejunal level, affecting its micellar solubility and further absorption, which could result in less fat absorption by body tissues involving the transactivation of several enzymes including lecithin cholesterol acyltransferase (LCAT) and lipoprotein lipase (LPL) [17,18]. 

The positive effect regulated by the combination of both strategies, systematic physical exercise and *S. maxima* supplementation, reveals a synergistic effect. In this sense, it has been described that on their own, exercise as well as *Spirulina* supplementation can modulate adaptations increasing heat-shock proteins and modulating muscle recovery [25] which often are not achieved individually in the short term, but in longer periods. In this sense, our systematic physical exercise program and *S. maxima* supplementation synergistically decrease BMI and increase VO_2_max, allowing the muscles to improve the ability to deal with high-intensity exercise stress in a short-term period [19].

The most relevant finding in our study are the synergistic effects of exercise and *S. maxima* intake on the blood lipid profile of dyslipidemic obese subjects and the correlations of the studied variables; however, whether these mechanisms indeed happen and what is their relationship with oxidative stress and physiological antioxidant response warrant future studies to understand the mechanisms involved in the general fitness improvements found. 

## 5. Conclusions

A systematic physical exercise program and/or *S. maxima* supplementation showed a beneficial reduction in BMI, VO_2_max, and blood lipid profile (TC, TAG, and LDL-C) of dyslipidemic obese men, and their effects on the body composition, blood lipid profile, and cardiorespiratory fitness, in overweight and obese men were correlated. 

## Figures and Tables

**Figure 1 antioxidants-08-00507-f001:**
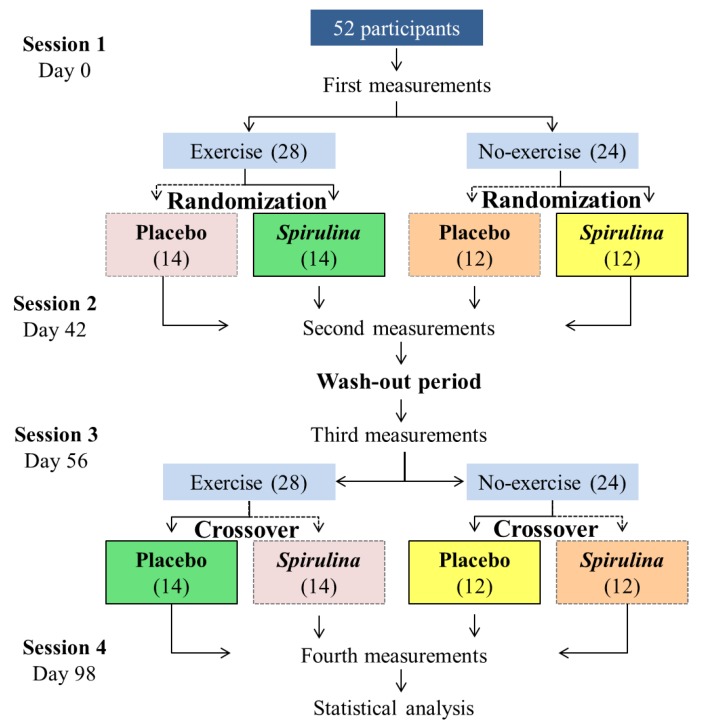
Experimental design for the effect of exercise and *Spirulina maxima* supplementation. The same color means the same group of participants.

**Table 1 antioxidants-08-00507-t001:** Baseline characteristics of participants.

	Total	Overweight	Obesity
*N*	52	27	25
Age (y)	26 ± 5	26 ± 4	27 ± 6
Bodyweight (kg)	90 ± 13	81 ± 8	100 ± 12
Height (m)	1.72 ± 0.1	1.72 ± 0.1	1.73 ± 0.1
BMI (kg·m^−2^)	30.2 ± 4	27.4 ± 1.2	33.3 ± 3.8
Body fat (%)	28.8 ± 7.2	24.8 ± 5.9	33.2 ± 6.1
Energy intake (kcal·day^−1^)	2054 ± 104	1977 ± 139	2054 ± 151
VO_2_max (mL·min^−1^·kg^−1^)	35.4 ± 6.9	39.6 ± 5.1	30.8 ± 5.6
Total cholesterol (mg·dL^−1^)	196 ± 36	177 ± 28	218 ± 30
Triacylglycerols (mg·dL^−1^)	142 ± 41	136 ± 35	150 ± 46
LDL-C (mg·dL^−1^)	134 ± 36	115 ± 27	158 ± 31
HDL-C (mg·dL^−1^)	34 ± 9.6	35.3 ± 8.2	32.5 ± 10.9

Data are expressed as mean ± standard deviation (SD). *N*: sample size, BMI: body mass index, VO_2_max: maximal oxygen uptake, LDL-C: low-density lipoprotein-cholesterol, HDL-C: high-density lipoprotein-cholesterol.

**Table 2 antioxidants-08-00507-t002:** Effect of treatments on the blood lipid profile, body composition, and cardiorespiratory fitness in the dyslipidemic obese subjects.

Variable	Basal	Final	*p*-Value
TC (mg·dL^−1^)	218 ± 30	184 ± 33	0.000
TAG (mg·dL^−1^)	150 ± 46	127 ± 35	0.007
LDL-C (mg·dL^−1^)	158 ± 31	122 ± 34	0.000
HDL-C (mg·dL^−1^)	32.5 ± 10.9	38.6 ± 9.6	0.004
BMI (kg·m^−2^)	33.3 ± 3.8	30.1 ± 4.9	0.003
VO_2_max (mL·min^−1^ kg^−1^)	30.8 ± 5.6	34.7 ± 6.2	0.003
BF (%)	33.2 ± 6.0	30.5 ± 7.1	0.050

Data are expressed as mean ± SD. TC: total cholesterol, TAG: triacylglycerols; LDL-C: low-density lipoprotein-cholesterol, HDL-C: high-density lipoprotein-cholesterol, BMI: body mass index, VO_2_max: maximal oxygen uptake, BF: body fat.

**Table 3 antioxidants-08-00507-t003:** Effect of treatments on the blood lipid profile, BMI, body composition and VO_2_max.

Variable	Systematic Physical Exercise Program (*N* = 28)					
	*S. maxima* Supplementation			Placebo Supplementation		
	Basal	Final	*p*-Value	Basal	Final	*p*-Value
BMI (kg·m^−2^)	29.6 ± 3	28.9 ± 3	0.359	29.7 ± 3	29.4 ± 3	0.755
Body fat (%)	26.4 ± 7	25.2 ± 6	0.493	27.0 ± 7	26.0 ± 7	0.592
TC (mg·dL^−1^)	196.3 ± 35	162.6 ± 33	0.001	197.0 ± 38	177.2 ± 36	0.053
TAG (mg·dL^−1^)	157.3 ± 47	134.8 ± 37	0.070	139.3 ± 44	124.2 ± 42	0.212
LDL-C (mg·dL^−1^)	127.9 ± 36	93.3 ± 34	0.001	137.4 ± 38	115.4 ± 36	0.046
HDL-C (mg·dL^−1^)	34.4 ± 9	41.7 ± 10	0.005	31.4 ± 9	36.2 ± 9	0.059
VO_2_max (mL·min^−1^ kg^−1^)	37.0 ± 7	39.5 ± 7	0.212	36.6 ± 8	38.2 ± 7	0.486
**Variable**	**No exercise program (*N* = 24)**					
	***S. maxima* Supplementation**			**Placebo Supplementation**		
	**Basal**	**Final**	***p*-Value**	**Basal**	**Final**	***p*-Value**
BMI (kg·m^−2^)	30.8 ± 5	30.2 ± 5	0.709	30.9 ± 5	30.8 ± 5	0.945
Body fat (%)	31.3 ± 7	30.0 ± 7	0.551	31.2 ± 7	30.9 ± 7	0.917
TC (mg·dL^−1^)	201.2 ± 39	183.0 ± 37	0.103	190.2 ± 31	185.7 ± 31	0.619
TAG (mg·dL^−1^)	141.4 ± 37	127.3 ± 34	0.176	131.2 ± 31	125.1 ± 32	0.517
LDL-C (mg·dL^−1^)	138.0 ± 37	117.2 ± 38	0.062	131.3 ± 32	127.4 ± 30	0.680
HDL-C (mg·dL^−1^)	34.9 ± 10	40.3 ± 11	0.071	35.3 ± 12	36.7 ± 8	0.655
VO_2_max (mL·min^−1^ kg^−1^)	33.6 ± 6	35.4 ± 7	0.399	33.7 ± 6	34.2 ± 6	0.796

Data are expressed as mean ± SD. BMI: body mass index, VO_2_max: maximal oxygen uptake, TC: total cholesterol, TAG: triacylglycerols; LDL-C: low-density lipoprotein-cholesterol, HDL-C: high-density lipoprotein-cholesterol; *N*: sample size.

**Table 4 antioxidants-08-00507-t004:** Correlation matrix among variables.

	Exercise and *Spirulina maxima* Supplementation					
	ΔBMI	ΔBF	ΔVO_2_max	ΔTC	ΔTAG	ΔLDL-C
Δ%BF	0.097	1				
ΔVO_2_max	−0.492 *	−0.217	1			
ΔTC	0.348	−0.164	0.091	1		
ΔTAG	0.162	0.101	−0.157	0.094	1	
ΔLDL-C	0.255	−0.164	0.035	0.798 *	−0.184	1
ΔHDL-C	−0.137	0.038	0.159	−0.2282	0.066	−0.690 *
	**Exercise and placebo Supplementation**					
	**ΔBMI**	**ΔBF**	**ΔVO_2_max**	**ΔTC**	**ΔTAG**	**ΔLDL-C**
Δ%BF	−0.094	1				
ΔVO_2_max	−0.151	−0.056	1			
ΔTC	0.329	−0.401 *	−0.246	1		
ΔTAG	0.030	−0.155	0.075	0.461 *	1	
ΔLDL-C	0.398 *	−0.280	−0.184	0.718 *	0.189	1
ΔHDL-C	0.021	0.028	−0.164	0.288	0.184	−0.286
	***Spirulina maxima* supplementation without exercise**					
	**ΔBMI**	**ΔBF**	**ΔVO_2_max**	**ΔTC**	**ΔTAG**	**ΔLDL-C**
Δ%BF	0.210	1				
ΔVO_2_max	−0.526 *	−0.497 *	1			
ΔTC	0370	0.160	−0.314	1		
ΔTAG	0.425 *	0.299	−0.103	0.243	1	
ΔLDL-C	0.135	0.237	−0.161	0.673 *	0.143	1
ΔHDL-C	0.131	−0.180	−0.059	0.297	0.041	−0.371

Δ = Changes between before and after values, BW: body weight, %BF: percentage of body fat, VO_2_max: maximal oxygen uptake, TC: total cholesterol, TAG: triacylglycerols; LDL-C: low-density lipoprotein-cholesterol, HDL-C: high-density lipoprotein-cholesterol. Asterisk (*) means *p* < 0.01.

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
