# Peer review of "Systematic Physical Exercise and Spirulina maxima Supplementation Improve Body Composition, Cardiorespiratory Fitness, and Blood Lipid Profile: Correlations of a Randomized Double-Blind Controlled Trial"

_antioxidants, 2019, doi:10.3390/antiox8110507_

Round 1

Reviewer 1 Report

They pretend to evidence that a systematic physical exercise program and Spirulina maxima (S. maxima) intake in overweight and obese men has beneficial effects on the body composition, blood lipid profile and cardiorespiratory fitness, and those effects are highly correlated. They design a double blind crossover study in overweight and obese subjects supplemented with placebo or S maxima. A half of them follow a systematic physical exercise program, whereas the other half no additional exercise program was followed. The number of obese subjects or overweight subjects in each group was indicated.

The study design not have into account the diet composition and energy intake of the two study groups at the beginning and also during the training program. This information is needed in order to attribute the effects observed in lipid profile and body composition in the systematic physical exercise program and s maxima supplementation.

Materials and methods

Lines 81- 82: ”… received randomly 4.5 g·d-1 of S. maxima or placebo (saccharine) in to 42-day crossover periods (randomly), separated by 14 days (wash-out period) (Figure 1)” The composition of placebo need to be more detailed. Really, does the placebo group intake 4.5g/day of saccharin?

Lines 96-97 how did you calculate the individual heart rate reserve in order to plan the exercise protocol?

The methods for blood sampling and processing and the analytical methods are not described.

Results

Line 114. Which were the cut off values used to declare the subjects with dyslipidemia?

Lines 119-121. Can the group with exercise and supplementation with S. maxima be considered without dyslipidemia?

Lines 119-1212. Explain more detailed the results obtained.

Discussion

The results obtained need to be more and better discussed.

Conclusion

The correlation detailed in Table 3 did not evaluate the effects of the systematic physical exercise program and S maxima supplementation. The conclusion  of ‘a systematic physical exercise program and S. maxima supplementation showed beneficial highly correlated effects on the body composition, blood lipid profile, and cardiorespiratory fitness, in overweight and obese men’ can’t be attained from these results.

Author Response

We truly appreciate your contributions. The changes requested can be identified in the new version of our manuscript (antioxidants-601408-R1) while a point-by-point response to your requests is described below:

They pretend to evidence that a systematic physical exercise program and Spirulina maxima (S. maxima) intake in overweight and obese men has beneficial effects on the body composition, blood lipid profile and cardiorespiratory fitness, and those effects are highly correlated. They design a double blind crossover study in overweight and obese subjects supplemented with placebo or S maxima. A half of them follow a systematic physical exercise program, whereas the other half no additional exercise program was followed. The number of obese subjects or overweight subjects in each group was indicated.

The study design not have into account the diet composition and energy intake of the two study groups at the beginning and also during the training program. This information is needed in order to attribute the effects observed in lipid profile and body composition in the systematic physical exercise program and s maxima supplementation.

R=Thanks for your comment. Your observation has been taken into account, so the diet composition and energy intake information has been added to the Methods Section (2.5) and the main outcomes of energy intake has been added to the Results Section.

Materials and methods

Lines 81- 82: ”… received randomly 4.5 g·d-1 of S. maxima or placebo (saccharine) in to 42-day crossover periods (randomly), separated by 14 days (wash-out period) (Figure 1)” The composition of placebo need to be more detailed. Really, does the placebo group intake 4.5g/day of saccharin?

R= More information has been added about supplements characteristics in Section 2.1, including the following statement: “Placebo composition consisted of a low-calorie saccharine powder. Both supplements were encapsulated in dark capsules (0.5 g each one) to present the same organoleptic characteristics, and subjects were asked to ingest three capsules before each daily meal. Due the lack of information in the literature, dose (4.5 g·day-1), duration for the treatment period (six weeks) and the washing period (two weeks) were considered long enough according to a systematic review of clinical trials that used Spirulina as treatment”.

Lines 96-97 how did you calculate the individual heart rate reserve in order to plan the exercise protocol?

R= The following paragraph in Section 2.7 about the determination of individual heart rate reserve has been added: “Target heart rate for each exercise session was based on a percentage of heart rate reserve (HRR). The difference between the maximal heart rate observed during the stress test at maximal intensity and the heart rate prior the test, measured with subjects resting comfortable for at least 15 min in a supine position (resting heart rate) was considered the HRR. The cardiovascular exercise was administered as follows: Three days at low to moderate intensity (50-80% of their HRR), and two days at high intensity (80-90% of their HRR)”. Finally, he stress test at maximal intensity was performed to determine the VO2max, and is described in Section 2.6.

The methods for blood sampling and processing and the analytical methods are not described.

R= Information about blood sample collection and biochemical analysis has been added in Section 2.8: “Each session before the stress test at maximal intensity, fasting blood samples (8mL, 8-10 h of fasting) of subjects were collected from the antecubital vein into ethylene diamine tetra acetic acid (EDTA) tubes (four in total, day 0, 42, 56 and 98). Samples were centrifugated at 4°C at 3,000 g during 20 min to obtain plasma, and the lipid profile (TC, TAG, LDL-C, and HDL-C) levels were analyzed by standard enzymatic procedures with a spectrophotometer at 505 nm.

Results

Line 114. Which were the cut off values used to declare the subjects with dyslipidemia?

R= The requested information has been added in the Section 2.8: “Abnormal cutoffs for TC, TAG, LDL-C and HDL-C were >200, >150, >100, and <40 mg·dL−1, respectively”.

Lines 119-121. Can the group with exercise and supplementation with S. maxima be considered without dyslipidemia?

R= Actually, dyslipidemic subjects were divided randomly in all treatment groups, but after the clinical trial only the HDL-C of obese subjects remained in dyslipidemic levels, the complete statement regarding this has been added to Results Section: “After the study, obese subjects showed statistical differences (p<0.01, basal vs. final) in BMI (33.3±3.8 vs. 30.1±4.9 kg·m−2), VO2max (30.8±5.6 vs. 34.7±6.2 mL·min−1·kg−1), and blood lipids (mg·dL−1): TC (218±30 vs. 184±33), TAG (150±46 vs. 127±35), LDL-C (158±31 vs. 122±34), and HDL-C (32.5±10.9 vs. 38.6±9.6), this means that only dyslipidemic HDL-C levels remained after the clinical trial”.

Lines 119-1212. Explain more detailed the results obtained.

R= Your suggestion has been considered, so the Results Section has been detailed more clearly, thank you.

Discussion

The results obtained need to be more and better discussed.

R= Results and Discussion Sections have been expanded to make clearer the main findings and scope of this clinical study. Thanks for your observation.

Conclusion

The correlation detailed in Table 3 did not evaluate the effects of the systematic physical exercise program and S maxima supplementation. The conclusion of ‘a systematic physical exercise program and S. maxima supplementation showed beneficial highly correlated effects on the body composition, blood lipid profile, and cardiorespiratory fitness, in overweight and obese men’ can’t be attained from these results.

R= According to you comment, we have rearranged Table 3, where the correlations of the effect of the treatments is shown separately to support our conclusions, we really appreciate your comments to improve the quality of our manuscript.

Sincerely,

Dr. Arnulfo Ramos-Jiménez, Corresponding author

Reviewer 2 Report

Summary

This manuscript showed that systematic physical exercise and Spirulina supplementation synergistically improved blood lipid profile in young sedentary people. Therefore, both systematic physical exercise and Spirulina supplementation might be promising prevention for cardiovascular disorders. So, I would recommend it for acceptance in Antioxidants after the major points listed below.

Major comments

The authors used Spirulina supplementation at 4.5 g/day. Why did you choose this amount for this experiment? Have you done preliminarily experiments?

Systematic physical exercise and Spirulina supplementation actually improved blood lipid profile in young sedentary people because Spirulina supplementation has several bioactive substances. Do you think what type of tissues or cells are important to improve lipid profile? Skeletal muscle, white adipocyte, brown adipocyte, and so on? This is a very important point in this study. Because only a systematic physical exercise or a Spirulina supplementation showed no effects on blood lipid profile, respectively.

The authors are submitting this manuscript to Antioxidants. However, there are no data about oxidative stress such as reactive oxygen species and lipid oxidation in this study. It might be a good idea to show data about oxidative stress. Did you examine lipid oxidation or contents of antioxidants in blood?

There were participants with obesity in this study. How about levels of glucose in blood of them? Because they might be going to be diabetes in the future.

Minor comments

Paragraph of “Result” is a very short compared with that of “Discussion”. You should write more result sentences.

When did participants take Spirulina supplementation during a day?

Author Response

We truly appreciate your contributions. The changes requested can be identified in the new version of our manuscript (antioxidants-601408-R1) while response to your requests is described below:

Summary

This manuscript showed that systematic physical exercise and Spirulina supplementation synergistically improved blood lipid profile in young sedentary people. Therefore, both systematic physical exercise and Spirulina supplementation might be promising prevention for cardiovascular disorders. So, I would recommend it for acceptance in Antioxidants after the major points listed below.

R= Thank you.

Major comments

The authors used Spirulina supplementation at 4.5 g/day. Why did you choose this amount for this experiment? Have you done preliminarily experiments?

R= According to your observation, more information has been added about supplements characteristics in Section 2.1, including the following statement: “Due the lack of information in the literature, dose (4.5 g·day-1), duration for the treatment period (six weeks) and the washing period (two weeks) were considered long enough according to a systematic review of clinical trials that used Spirulina as treatment”.

Systematic physical exercise and Spirulina supplementation actually improved blood lipid profile in young sedentary people because Spirulina supplementation has several bioactive substances. Do you think what type of tissues or cells are important to improve lipid profile? Skeletal muscle, white adipocyte, brown adipocyte, and so on? This is a very important point in this study. Because only a systematic physical exercise or a Spirulina supplementation showed no effects on blood lipid profile, respectively.

R= Thank you for your comment. We argue that the positive effect regulated by the combination of both strategies, systematic physical exercise and S. maxima supplementation, reveals a synergistic effect. In this sense, it has been described that on their own, exercise as well as Spirulina supplementation can modulate adaptations increasing heat-shock proteins and modulating muscle recovery which often are not achieved individually in the short term, but in longer periods In this sense, our systematic physical exercise program and S. maxima supplementation synergistically decrease BMI and increase VO2max, allowing the muscles to improve the ability to deal with high intensity exercise stress in a short-term period. This information has been added to the manuscript Discussion Section.

The authors are submitting this manuscript to Antioxidants. However, there are no data about oxidative stress such as reactive oxygen species and lipid oxidation in this study. It might be a good idea to show data about oxidative stress. Did you examine lipid oxidation or contents of antioxidants in blood?

R= Thank you for your comment. The antioxidant profile in plasma was examined, however, it was not possible to obtain complete data due to the poor or null detection of the markers. The next paragraph has been added to the Discussion Section: “Normally, oxidative species are maintained within healthy limits in order to prevent undesirable events (e.g. oxidative stress). However, during high physical activity reactive oxygen/nitrogen species are produced as messengers that regulate several physiological adaptations such as angiogenesis, insulin sensitivity, hypertrophy, mitochondrial biogenesis, among others that promotes tolerance to future stressors. In this regard, some studies have evaluated the influence of antioxidant supplementation on alterations induced by physical training, being the results obtained extremely variable. For example, it has been reported that supplementation with Vitamin C and Vitamin E during exercise decrease the endogenous antioxidant response and in other cases it has a null effect on mitochondrial hormesis”.

There were participants with obesity in this study. How about levels of glucose in blood of them? Because they might be going to be diabetes in the future. R= Glucose levels were taken before the beginning of the study to ensure subjects did not present diabetes. The inclusion criteria have been added in Section 2.3, including the following statement “Male volunteer, sedentary (daily energy expenditure <4 metabolic equivalents), with BMI≥25 kg·m−2 and 18-35 years old were recruited in an intra-school campaign and personalized interviews to ensure eligibility. Exclusion criteria were drinking more of 100 mL of alcohol a week, taking drugs or diet supplements, presenting diabetes or any other chronic disease, and having an impediment to practice regular physical exercise”. Also, glucose levels were taken during each stress test at maximal intensity to check the subject’s health during each test (Information added in section 2.6).

Minor comments

Paragraph of “Result” is a very short compared with that of “Discussion”. You should write more result sentences.

R= Results and Discussion Sections have been expanded to make clearer the main findings and scope of this clinical study. We appreciate your observations to improve the quality of our manuscript.

When did participants take Spirulina supplementation during a day?

R= According to your comment, supplements encapsulation and ingest information was added to Section 2.1, including the following statement: “Both supplements were encapsulated in dark capsules (0.5 g each one) to present the same organoleptic characteristics, and subjects were asked to ingest three capsules before each daily meal.”.

Sincerely,

Dr. Arnulfo Ramos-Jiménez, Corresponding author

Round 2

Reviewer 2 Report

I would like to accept this revised manuscript for Antioxidants.

Author Response

October 17th, 2019

Thank you for the time devoted to our article entitled: Systematic physical exercise and Spirulina maxima supplementation improve body composition, cardiorespiratory fitness, and blood lipid profile: Correlations of a randomized double-blind controlled trial, authored by Marco Antonio Hernández-Lepe, Francisco Javier Olivas-Aguirre, Luis Mario Gómez-Miranda, Rosa Patricia Hernández-Torres, José de Jesús Manríquez-Torres, and Arnulfo Ramos-Jiménez (corresponding author). We have answered the minor comments in the new version of our manuscript (antioxidants-601408-R2) according to your questions. Corrections and modifications were highlighted in red along the manuscript.          

Thanks for your attention , sincerely,

Dr. Arnulfo Ramos-Jiménez, Corresponding author